

# Response relationship between atmospheric O₃ and its precursors in Beijing based on smog chamber simulation and a revised MCM model

Jialin Lu[1,2], Tianzeng Chen[1,2][*], Jun Liu[1,2], Huiying Xuan[1,2], Peng Zhang[1,2], Qingxin Ma[2,3], Yonghong Wang[1,2], Hao Li[1,2], Biwu Chu[2,3,*], Hong He[1,2,4]

[1]Laboratory of Atmospheric Environment and Pollution Control, Research Center for Eco-Environmental Sciences, Chinese Academy of Sciences, Beijing 100085, China

[2]University of Chinese Academy of Sciences, Beijing 100049, China

[3]State Key Laboratory of Regional Environment and Sustainability, Research Center for Eco-Environmental Sciences, Chinese Academy of Sciences, Beijing, 100085, China

[4]State Key Laboratory of Advanced Environmental Technology, Institute of Urban Environment, Xiamen 361021, China

*Correspondence to*: tzchen@rcees.ac.cn (Tianzeng Chen) and bwchu@rcees.ac.cn (Biwu Chu)

**Abstract.** Ozone (O₃) pollution has been receiving increasing attention, but its simulation performance in models remains unsatisfactory. This study characterized the response relationship between O₃ and its precursors in the atmospheric relavant condition through a combination of smog chamber experiments and Master Chemical Mechanism (MCM) box model. By

adding chamber wall related reaction mechanisms, the model achieved significant improvement in simulating O₃ with an Normalized Mean Bias (NMB) value changing from -76.1 % to -12.7 %. The enhanced model was subsequently extended to the ambient atmospheric conditions in the Daxing District of Beijing, incorporating the parameterization of ground related reactions, heterogeneous reactions of Nitrogen Dioxide (NO₂), and unidentified NO₂ sinks. Compared to basic model, the resulting revised model demonstrated substantially enhanced accuracy in simulating ambient O₃ concentrations with an

Normalized Mean Bias (NMB) value changing from 113.8 % to -5.2 % and enhanced O₃ formation sensitivity to Volatile Organic Compounds (VOCs) in Daxing District. These findings underscore that incorporating interface mediated chemical processes and accounting for unidentified NO₂ sinks into model is critical for determining the sensitivity of O₃ formation and optimizing regional O₃ pollution control strategies.

## 1 Introduction

Ozone (O₃), a secondary air pollutant, adversely impacts natural vegetation, agricultural crops, and human health (Feng et al., 2019; Lefohn et al., 2018). Despite China's implementation of stringent air pollution mitigation measures over the past decade (2014–2023), which reduced PM₂.₅ concentrations in the Beijing-Tianjin-Hebei region by approximately 35 %, the 90th percentile of maximum daily 8-hour average O₃ concentrations (MDA8-O₃) have persistently stayed around of 180 μg m⁻³ (https://www.mee.gov.cn/). This concentration exceeds the safe limit of 100 μg m⁻³ which was 99th percentile of MDA8-O₃

and recommended by the World Health Organization (https://www.who.int/news-room/feature-stories/detail/what-are-the-



who-air-quality-guidelines). China is facing an increasingly prominent $O_3$ pollution problem. Developing effective strategies to mitigate $O_3$ pollution has become one of the most pressing environmental challenges in China (Wang et al., 2020).

$O_3$ formation is primarily associated with two precursors: VOCs and $NO_x$ (Haagen-Smit, 1952). The fundamental pathways of $O_3$ formation comprise three sequential stages: (1) atmospheric oxidation of VOCs generates peroxy radicals ($RO_2$ and $HO_2$

radicals); (2) these radicals react with NO to form $NO_2$; and (3) photolytic decomposition of $NO_2$ produces $O_3$ (Bozem et al., 2017; Pusede et al., 2015). Considering the complex nonlinear relationship between $O_3$ formation and its precursors, $O_3$ control remains a persistent challenge in atmospheric environment. The sensitivity range of $O_3$ formation can be divided into three distinct regimes: (1) $NO_x$-limited regime, (2) VOC-sensitive regime, and (3) transitional regime with mixed precursors influence (Chu et al., 2024). However, the complex of atmospheric conditions hinders accurate characterization of chemical

processes in models, resulting in significant biases in sensitivity analysis of $O_3$ formation (Li et al., 2018; Ma et al., 2021; Qu et al., 2021; Chen et al., 2024), and triggering debates over optimal precursor control strategies. Therefore, it is crucial to investigate the key factors influencing $O_3$ formation to provide a scientific basis for the prevention and control of $O_3$ pollution. Recent modeling studies have demonstrated that parameterization of ground mediated chemical processes enhances the simulation accuracy of $O_3$ production (Qin et al., 2025; Zhang et al., 2019). However, existing studies have predominantly

focused on the heterogeneous conversion processes of $NO_2$ (Qin et al., 2025; Zhang et al., 2019), while the potential contributions of other ground mediated chemical reactions to $O_3$ production remain systematically unassessed probably due to the complex and diverse ground types.

Smog chamber has emerged as an indispensable approach for studying how secondary pollutants like $O_3$ formation (Chen et al., 2022; Pierce et al., 1995b). The smog chamber can simulate ambient atmospheric conditions under controlled

meteorological settings. This allows for the study of physicochemical reactions while minimizing interference from meteorological variables. However, the chamber walls are not completely inert surfaces. Heterogeneous reactions may occur on these surfaces, accompanied by interfacial physical processes such as gas and aerosol adsorption/desorption (Pinho et al., 2005; Killus and Whitten, 1990; Chu et al., 2021), which will introduce biases into experimental data. Thus, it is necessary to consider the additional physicochemical mechanisms mediated by the chamber walls (wall effects) when analyzing chamber

data.

The establishment of the Master Chemical Mechanism (MCM) relies heavily on smog chamber experiments (Wyche et al., 2010; Bloss et al., 2005), and its model systems serve as classical tools for simulating atmospheric $O_3$ formation (Shek et al., 2022; Wang et al., 2018; Liu et al., 2022). Toluene, as a typical anthropogenic VOC (Li et al., 2020) and isoprene, as a representative biogenic VOC (Guenther et al., 1995), both have high $O_3$ formation potential (Derwent et al., 1998). In our

study, the smog chamber experimental results of toluene and isoprene mixed precursor systems were used as constraints for the MCM model to parametrize the wall effects allowing for the derivation of a revised model. And then, the influence of ground mediated reactions on the formation of $O_3$ was systematically explored by extrapolating the revised MCM model to



ambient atmospheric scenarios. Additionally, the simulation performance for $O_3$ in field was assessed by adjusting the ground

surface heterogeneous reaction rate of $NO_2$, incorporating heterogeneous reactions of $NO_2$ on aerosol surfaces, and introducing

an unknown sink for $NO_2$ within the model. The findings provide crucial theoretical foundation for precisely formulating

prevention and control strategies targeting regional atmospheric $O_3$ pollution.

## 2 Experimental methods

The specific configuration and working principles of our 30 $m^3$ smog chamber system have been clearly described in our

previous studies (Chen et al., 2019b; Chen et al., 2019a). To be brief, the rectangular chamber was made of FEP Teflon film

with a thickness of 125 μm. A magnetic levitation fan is installed at the center of the bottom of the chamber to mix the reactants.

The outside of the chamber is surrounded by stainless steel mirror panels, which are used to reflect ultraviolet light and make

the irradiance uniform inside the chamber. One hundred and twenty ultraviolet lamps (Philips TL 60/10R) with a peak intensity

of 365 nm are embedded in the stainless steel panels. After all the ultraviolet lamps are turned on, the photolysis rate of $NO_2$,

used to characterize light intensity, was experimentally determined to be 0.0092 $s^{-1}$ (Chen et al., 2022). And this is comparable

to the light intensity at noon in Beijing. Before starting the experiment, the chamber was flushed with zero air at a flow rate of

100 L $min^{-1}$ until the concentration of gas- and particle-phase contaminants is sufficiently low. An air conditioner is installed

outside the chamber to control the reaction temperature, with an accuracy of ±1 ℃.

### 2.1 Smog chamber experiments

Toluene, isoprene and $NO_x$ reserved in gas cylinders were introduced into the chamber using a mass flow controller. By

precisely controlling the introduction time and flow rate, we managed to obtain precursor systems with different concentrations.

The concentrations of toluene and isoprene were measured by a Vocus proton transfer reaction time-of-flight mass spectrometer

(Vocus PTR-ToF-MS, Tofwerk AG, Aerodyne Research). The concentrations of $NO_x$ and $O_3$ were measured using the

NO−$NO_2$−$NO_x$ analyzer (model 42i-TL, Thermo) and the $O_3$ analyzer (model 49i, Thermo), respectively.

**Table 1: Detailed experimental conditions inside the smog chamber.**

| exp. no.[1] | RH (%) | T (℃) | $Isoprene_0$ (ppb) | $Toluene_0$ (ppb) | $NO_0$ (ppb) | $NO_{2,0}$ (ppb) | $NO_{x,0}$ (ppb) | (Isoprene+Toluene)/$NO_{x,0}$ (ppbC $ppb^{-1}$) | ΔIsoprene (ppb) | ΔToluene (ppb) | $ΔO_3$ (ppb) |
|---|---|---|---|---|---|---|---|---|---|---|---|
| Iso&Tol01 | 53−62 | 28−30 | 11.52 | 11.31 | 25.08 | 0 | 24.58 | 5.56 | 9.82 | 2.14 | 45.56 |
| Iso&Tol02 | 52−61 | 28−30 | 2.89 | 10.12 | 25.46 | 0 | 24.82 | 3.44 | 1.96 | 2.56 | 28.89 |
| Iso&Tol03 | 53−62 | 28−30 | 11.70 | 1.43 | 26.06 | 0 | 25.03 | 2.74 | 9.90 | 0.12 | 30.02 |





| Iso&Tol04 | 52–59 | 29–30 | 10.68 | 10.80 | 5.03 | 0.03 | 5.06 | 25.51 | 8.93 | 3.86 | 64.44 |
| Iso&Tol05 | 52–61 | 28–30 | 8.26 | 6.31 | 1.01 | 0.27 | 1.28 | 66.78 | 7.30 | 1.60 | 51.61 |
| Tol01 | 53–61 | 28–30 | $0^2$ | 10.95 | 25.38 | 0 | 24.61 | 3.18 | 0 | 4.10 | 22.95 |
| Iso01 | 53–61 | 28–30 | 10.92 | 0 | 25.79 | 0 | 25.04 | 2.32 | 8.72 | 0 | 30.87 |
| Iso02 | 52–64 | 27–30 | 13.16 | 0 | 26.29 | 0 | 25.43 | 2.70 | 10.89 | 0 | 34.93 |

[1]Iso represents isoprene, Tol represents toluene; [2]0 means that the relevant precursor was not added to the chamber.

Once all precursors concentration stabilized within the smog chamber, the fan was turned off, and the ultraviolet lamp was turned on to initiate the photochemical experiment, which persisted for a duration of 6 h. During the experiment, a temperature

and humidity probe (Vaisala HMP110, Finland) was used to measure the temperature and relative humidity inside the chamber with a time resolution of 1 min. Fig. S1 and S2 reflect that the relative humidity and temperature were precisely regulated and maintained within the desired ranges during the experiments. The detailed experimental parameters are described in Table 1. According to the initial concentration ratios of VOCs and $NO_x$, the experiments covered three sensitivity regimes for $O_3$ formation.

**2.2 The MCM box model (AtChem2)**

AtChem2 model is constructed based on MCM: an explicit chemical mechanism. Although it does not take into account complex meteorological parameters, owing to its zero dimensional box model structure, it contains relatively complete atmospheric chemical reactions and enables rapid numerical simulation of atmospheric chemical processes. The core operation process of the model comprises the following key steps. Firstly, the initial concentrations of reactants, constraint conditions,

and the chemical mechanism document are input. Subsequently, the coupled ordinary differential equations describing the chemical reactions in chemical mechanism document is compiled into Fortran executable code. Finally, the numerical integration algorithm for differential equations is employed to solve for the temporal evolution of the system variables. Detailed descriptions regarding the AtChem2 model have been reported in previous study (Cox et al., 2020).

In the smog chamber simulation, the $NO_2$ photolysis frequency, $J_{NO_2}$, was set to a constant value of 0.0092 s$^{-1}$, owing to the

stable optical environment within the chamber. The calculation method of photolysis rates are provided as shown in Eq. (1) (Goliff et al., 2004; Borrás et al., 2024). By comparing the relationship between the experimental and calculated values of $J_{NO_2}$, the values of other photolysis rates in the model can be deduced (Carter et al., 1995a).

$$J = \int I(\lambda)\sigma(\lambda)\phi(\lambda)d\lambda \ , \qquad\qquad\qquad (1)$$

In the Eq. (1), the actinic flux I is used to characterize the distribution of light intensity within the smog chamber. The

absorption cross section σ and the quantum yield Φ describe the molecular light absorbing properties and energy conversion

efficiency, respectively, during photolysis. The value of I is calculated based on the actual spectral data measured by the

Miniature Fiber Optic Spectrometer as shown in Fig. S3. The parameters of σ and Φ are sourced from the MCM database. The

final numerical settings of all the photolysis rate in the model are comprehensively presented in Table S1.

**2.3 Description of observation site**

The observation site is located on the campus of Beijing Institute of Petrochemical Technology in Daxing District, Beijing

(39.73° N, 116.33° E) and the observation period was from 11 August 2019 to 19 August 2019. The details information about

this site and measuring instruments can be found in our previous study (Chen et al., 2021; Xuan et al., 2023). Briefly, the

observation platform is set on the top of the teaching building, approximately 27 m above the ground vertically. And the

observation site is situated in the southern suburbs of Beijing, located in the northeastern part of the North China Plain. The

surrounding environment of the observation point includes residential areas, educational institutions, and urban arterial roads.

VOCs concentrations were quantified via vacuum ultraviolet single-photon ionization time-of-flight mass spectrometry

(SPIMS-3000, Guangzhou Hexin Analytical Instrument Co., Ltd., China). The concentration of $NO_x$, $SO_2$, $O_3$, and CO was

monitored using Thermo analyzers (Models 42i-TL, 43i, 49i, and 48i, respectively). HONO concentration was detected by a

wet-chemical long-path absorption photometer (WLPAP, Zhichen Beijing) and the particle surface area concentration was

derived from the scanning mobility particle sizer (SMPS, Model 3082 equipped with 3776 CPC, TSI Inc., USA). Additionally,

the meteorological parameters (temperature, relative humidity, wind speed, wind direction and pressure) were recorded by an

automatic weather station (Vaisala M451) and photolysis frequency of $NO_2$ ($J_{NO_2}$) was measured using a filter radiometer

(Metcon GmbH, Germany). The time series of several representative parameters are shown in Fig. S5.

**3 Results and discussion**

**3.1 Construction of a revised model accounting for chamber wall effects**

Figure 1 presents the evolution of $NO_x$, $O_3$ and each VOC concentrations during the photochemical reaction systems with the

mixed VOCs of toluene and isoprene. Two groups of typical experiments were selected to show the simulation results and

improvement. For the experiment of Iso&Tol02, the initial ratio of VOCs to $NO_x$ indicates that $O_3$ formation is under VOC-

limited regime, and during which the photochemical processes can characterize the typical urban atmospheric environment.

The experiment Iso&Tol04 is under $NO_x$-limited regime, corresponding to the atmospheric chemical characteristics of typical

rural areas (Cheng et al., 2019). As shown in Fig. 1, there is a huge deviation between the simulation results of the basic model

and the experimental values. For the Iso&Tol02 and Iso&Tol04 experiments, the $NO_2$ concentrations simulated by the basic

model are significantly lower than the measured values. Considering that toluene is primarily consumed by oxidation with OH

radicals, the degradation rate of toluene can reflect the concentration of OH radicals. However, the simulated degradation rates



of toluene by basic model are considerably lower than the measured values (Fig. 1e and 1j), suggesting that there is a huge gap

between the simulated and actual concentrations of OH radicals. For the Iso&Tol02 and Iso&Tol04 experiments, the $O_3$ values

simulated by the basic model are lower than the measured values, with NMB values of -83.9 % and -41.3 %, respectively. The

calculation method of NMB is detailed in Section S1. The significant discrepancy between simulations and measurements lies

in the model's failure to account for wall related reaction mechanisms. Identifying these reactions and incorporating their

parameterization into the model is therefore imperative.

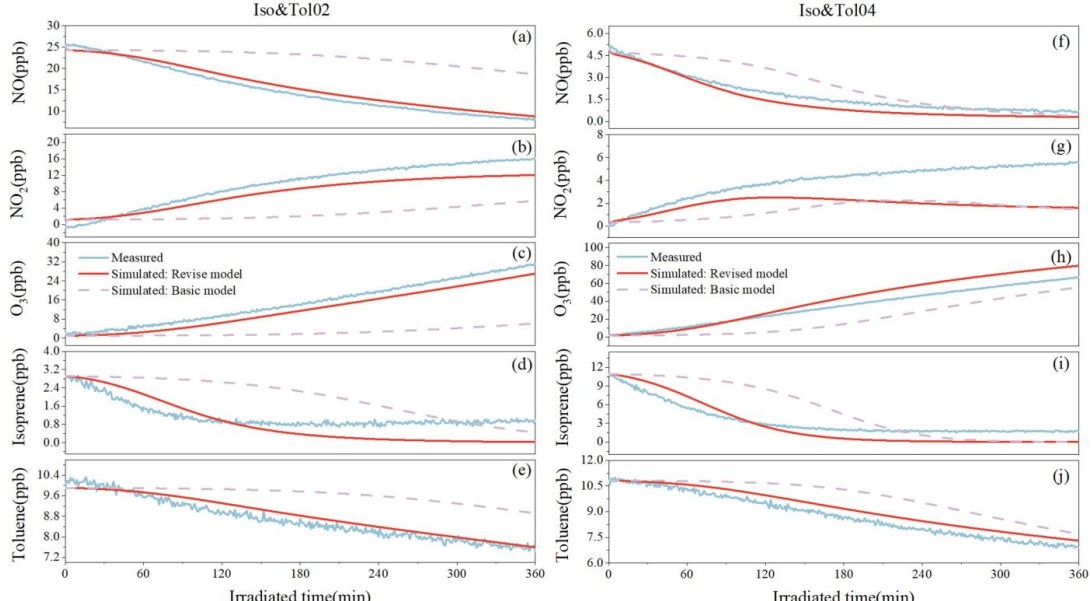

**Figure 1: Time series of (a, f) NO, (b, g) NO₂, (c, h) O₃, (d, i) isoprene and (e, j) toluene during the photochemical reaction process in experiments of Iso&Tol02 and Iso&Tol04. The blue line represents the measured values, the red line denotes the simulation results**

**of the revised model, and the dashed line indicates the simulation results of the basic model. The NMB values for O₃ simulated by the basic model, revised model in experiment Iso&Tol02 were -83.9 % and -19.0 %, respectively. And that in experiment Iso&Tol04 were -41.3 % and 20.0 %, respectively.**

Currently, numerous studies have investigated the chamber wall effects and determined the corresponding reaction rate

constants (Lurmann et al., 1991; Bloss et al., 2005; Metzger et al., 2008; Wang et al., 2014). Table 2 summarizes the reactions

related to the wall effects and their reaction rate constants used in our study. The wall photogenerated OH radicals mechanism

is incorporated into the model to bridge the discrepancy between simulated and measured values. The feasibility of this

mechanism has been verified in the studies conducted by Angove et al. (2005). and Lurmann et al. (1991). In order to better

simulate the degradation trend of VOCs, the optimal generation rate constant of OH radicals in the model is determined to be

$1.2 \times 10^6$ molecules $cm^{-3}$ $s^{-1}$, which is close to the findings of Wang et al. (2014). Meanwhile, the mechanism of light-induced



release of $NO_2$ from the wall (Bloss et al., 2005; lurmann et al., 1991) was introduced to address the issue of relatively low

production of $NO_2$ in the simulation. The determined optimal release rate constant of $NO_2$ is $6\times10^5$ molecules $cm^{-3}$ $s^{-1}$, falling

between the results reported by Angove et al. (2005) and Wang et al. (2014). In addition, Teflon wall can release small amounts

of organic impurities, which will consume OH radicals and generate $HO_2$ radicals (Metzger et al., 2008). Therefore, the

additional mechanism that converts OH radicals into $HO_2$ radicals was also introduced into the model. This mechanism can

accelerate the consumption of NO and also compensating for the deficiency of the simulated $NO_2$ and $O_3$ concentration. The

optimal conversion rate constant of OH radicals to $HO_2$ radicals is determined to be 10 $s^{-1}$, which lies within the range

mentioned by Lurmann et al. (1991).

**Table 2: The additional mechanisms related to wall effects and their associated reaction rate parameters.**

|  | Additional mechanisms | Parameter | Notes |
|---|---|---|---|
| 1 | $h\nu + wall \rightarrow OH$ | $1.2\times10^6$ molecule $cm^{-3}$ $s^{-1}$ | Refered to Angove et al. (2005) |
| 2 | $h\nu + wall \rightarrow NO_2$ | $6\times10^5$ molecule $cm^{-3}$ $s^{-1}$ | Refered to Angove et al. (2005) |
| 3 | $OH \rightarrow HO_2$ | 10 $s^{-1}$ | Refered to Lurmann et al. (1991) |
| 4 | $N_2O_5 \rightarrow 2wHNO_3$[1] | $1\times10^{-5}$ $s^{-1}$ | Adopted from Bloss et al. (2005) |
| 5 | $N_2O_5 + H_2O \rightarrow 2wHNO_3$ | $1\times10^{-20}$ $cm^3$ $s^{-1}$ | Adopted from Bloss et al. (2005) |
| 6 | $HNO_3 \rightarrow wHNO_3$ | $1\times10^{-4}$ $s^{-1}$ | Adopted from Bloss et al. (2005) |
| 7 | $NO_2 \rightarrow 0.5HONO + 0.5wHNO_3$ | $1\times10^{-6}$ $s^{-1}$ | Adopted from Angove et al. (2005) |
| 8 | $wHNO_3 + h\nu \rightarrow OH + NO_2$ | $J_{wHNO_3}$ | Theoretical calculation |
| 9 | $O_3 \rightarrow wall$ | $5.53\times10^{-6}$ $s^{-1}$ | Chamber characterization/measured |

[1]$wHNO_3$ represents adsorbed $HNO_3$ on the wall.

In addition, the heterogeneous reactions and homogeneous hydrolysis of $N_2O_5$, the wall losses of $HNO_3$ and $O_3$, the

photolysis of $wHNO_3$ as well as the heterogeneous reactions of $NO_2$ have also been considered as wall related reaction

mechanisms (Metzger et al., 2008; Bloss et al., 2005). These reactions need to be incorporated into the model. The rate

constants of the heterogeneous reaction and homogeneous hydrolysis of $N_2O_5$ are set at $1\times10^{-5}$ $s^{-1}$ and $1\times10^{-20}$ $cm^3$ $s^{-1}$,

respectively. The wall loss rate constant of $HNO_3$ is taken as $1\times10^{-4}$ $s^{-1}$. These values are consistent with the studies of Metzger

et al. (2008) and Angove et al. (2005). The rate constant of the heterogeneous reaction of $NO_2$ is set at $1\times10^{-6}$ $s^{-1}$, which is in

line with the research of Angove et al. (2005). The photolysis rate constant of $wHNO_3$ is assumed to be the same as that of

gaseous $HNO_3$ and is calculated to be $5.66\times10^{-7}$ $s^{-1}$ by the Eq. (1). The wall loss rate constant of $O_3$ is experimentally determined



to be $5.53 \times 10^{-6}$ s$^{-1}$. After introducing additional mechanisms, a revised model has been developed.

### 3.2 The performance evaluation of the revised model

As shown in Fig. 1, after applying the revised model, the simulation accuracy of each experimental parameter has been significantly improved. The NMB of O$_3$ in experiment of Iso&Tol02 increased significantly from -83.9 % to -19.0 %, and that

in experiment of Iso&Tol04 increased from -41.3 % to 20.0 %. The simulation performance of O$_3$ concentrations in other experiments was also significantly improved. In experiments of Iso&Tol01, Iso&Tol03, Iso&Tol05, Tol01, Iso01, Iso02, the NMB of O$_3$ changed from -91.1 % to -3.8 %, -83.9 % to -12.5 %, -56.9 % to -26.6 %, -81.9 % to -13.8 %, -84.4 % to -24.1 %, and -85.7 % to -21.6 %, respectively as shown in Fig. S4. Following the implementation of the revised model, the NMB of all experiments are distributed within the range of from -26.6 % to 20.0 %, with an average of -12.7 %. Consequently, the revised

model substantially reduced the discrepancies between simulated and observed values for key experiment parameters compared to the basic model, establishing a basis for subsequent sensitivity analysis.

### 3.3 Impact of the revised model on the sensitivity of O$_3$ formation in smog chamber compared to the basic model

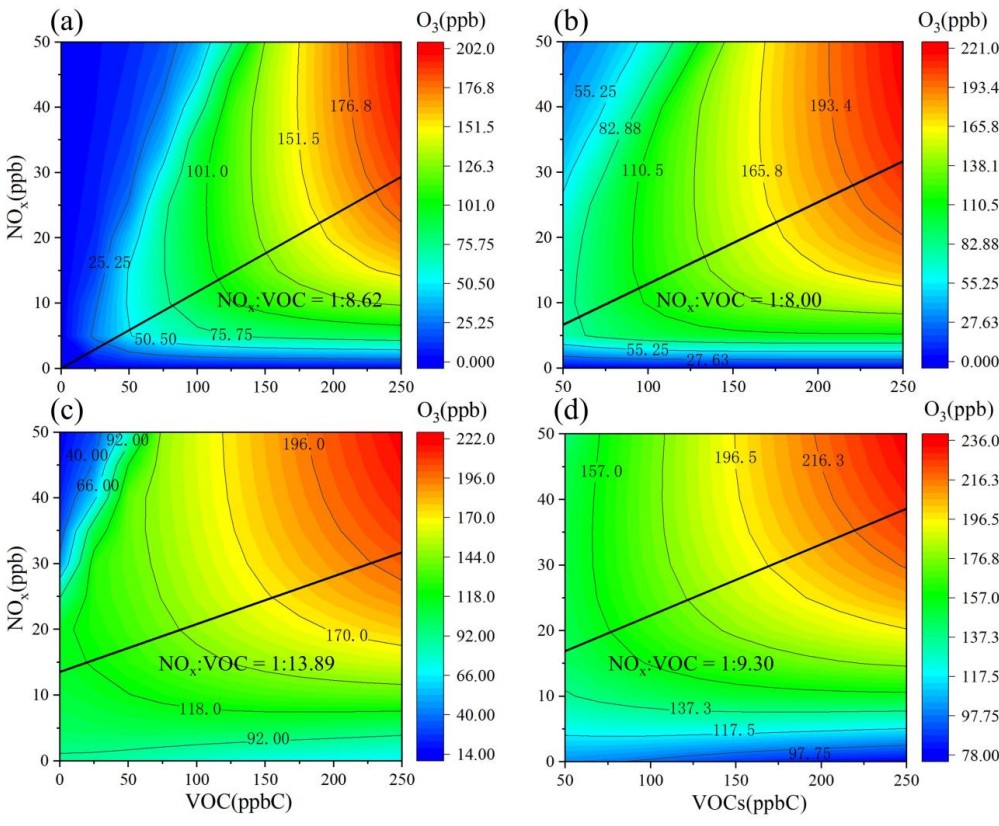

**Figure 2: Simulated EKMA curves for O$_3$ generation under the (a) toluene only system and (b) mixed VOCs system using the basic**



**model. Correspondingly, (c) and (d) display the simulated EKMA curves for O$_3$ generation under the toluene only system and mixed VOCs system, respectively using the revised model.**

In order to determine the specific impacts of the revised model on the sensitivity of O$_3$ formation, the EKMA (Empirical

Kinetic Modeling Approach) curves were obtained by simulating O$_3$ formation under varying concentration of VOCs and NO$_x$

(Liu et al., 2022; Ma et al., 2021). As illustrated in Fig. 2, the simulation results demonstrated the nonlinear response

characteristics of O$_3$ production to precursors. Meanwhile, it can be found that, whether in the toluene only system or in the

toluene isoprene mixed system, the slope of the ridge line of the EKMA curves derived from the revised model changes. This

indicates that the model revision alters the sensitivity of O$_3$ formation to its precursors. Therefore, it is necessary to incorporate

mechanisms related to wall effects into the model to accurately capture the O$_3$ formation sensitivity. Comparing scenarios with

and without isoprene (Fig. 2d and 2c), the results showed that the slope of the EKMA ridge line increases (1:13.89 vs. 1:9.30)

when isoprene are considered, further emphasizing the necessity of stringent NO$_x$ control for effective O$_3$ pollution mitigation.

Consequently, to establish a more accurate precursor response relationship, it is essential to account for the contribution of

biogenic VOCs (such as isoprene) in the model. This conclusion is also supported by the findings of Tan et al. (2018).


**3.4 The impact of the revised model that accounts for surface-to-volume ratio on the atmospheric O$_3$ formation compared to the basic model**

The applicability of the revised model under atmospheric conditions was further investigated. Due to the difference in surface-

to-volume ratio between the atmospheric environment and smog chamber, the rate constants related to surface reactions should

be adjusted when applying the revised model to the ambient atmosphere. To facilitate the study of atmospheric O$_3$ formation

sensitivity in a zero dimensional box model, the boundary layer height was assumed to be ranged from 300 m at night to 1500

m in the afternoon throughout the simulations (Gao et al., 2014; Xuan et al., 2024; Wang et al., 2025), and the atmospheric

surface-to-volume ratio was calculated using Eq. (3) (Li et al., 2010). Assuming that the uptake coefficient in the chamber wall

is equal to that in the atmospheric ground, the ground related reaction rate constant can be derived using Eq. (2), which presents

the calculation method for the heterogeneous reaction rate constant of HONO (Gao et al., 2014).

$$k = \frac{1}{8} \gamma v_{NO_2} \frac{S}{V_g} \, , \tag{2}$$

$$\frac{S}{V_g} = \frac{1.7}{H} \, , \tag{3}$$

where $k$ is defined as the surface reaction rate constant for HONO, $\gamma$ represents the uptake coefficient, $v_{NO_2}$ denotes the

average molecular velocity of NO$_2$, and S/V$_g$ stands for the surface-to-volume ratio of the ground. H denotes the boundary

layer height.




Specifically, the surface reaction rate constant in the ambient atmosphere at night and in the afternoon can be obtained by multiplying the corresponding values in the chamber by $2.9 \times 10^{-3}$ and $5.8 \times 10^{-4}$, respectively. In the further revised model, the surface reaction rate constants were adjusted in the manner described above to develop an atmospheric relevant revised model that accounts for surface-to-volume ratio, named SVR model. This parameterization strategy specifically addresses surface

mediated kinetic disparities between smog chamber and atmospheric environment.

In the SVR model and basic model, the observed species concentrations and meteorological parameters are constrained. Inorganic pollutants include $NO$, $SO_2$ and $CO$, while meteorological parameters encompass temperature, relative humidity, atmospheric pressure, and the photolysis rate of $NO_2$. Considering both the operational efficiency and the accuracy of simulations, the top 20 VOCs with the highest maximum incremental reactivity (MIR) values are selected for constraint (Carter,

2009). Detailed information on the constrained species can be found in Table S2. It should be noted that we did not constrain $NO_2$ and HONO. Additional mechanisms in Table 2 involve the source and sink processes of $NO_2$. Constraining $NO_2$ in the model will weaken the research significance of the revised model. HONO serves as a significant source of OH radicals in the atmospheric environment, exerting a critical influence on $O_3$ formation. Moreover, its source−sink processes are closely linked to $NO_x$, meaning that constraining HONO could hinder the analysis of $O_3$ reduction scenarios.


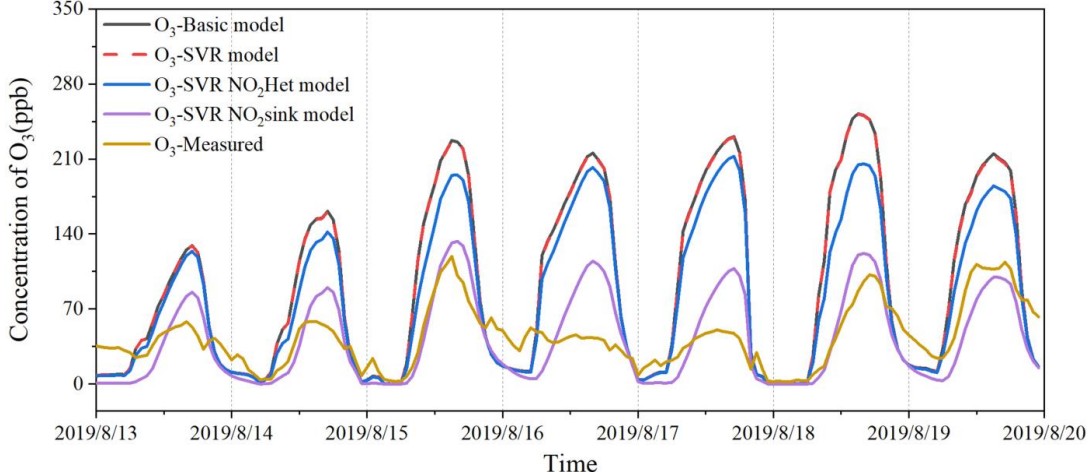

**Figure 3: Measured concentrations of O₃ and the simulation results of different models in Daxing, Beijing from 13−19 August 2019. The NMB values for O₃ simulated by the basic model, SVR model, SVR NO₂Het model, and SVR NO₂sink model were 113.8 %, 113.0 %, 84.1 % and -5.2 %, respectively.**


Unexpectedly, the simulation results of SVR model and basic model showed minimal differences (Fig. 3). It is speculated that this is due to the too small surface-to-volume ratio ($5.7 \times 10^{-3}$–$1.1 \times 10^{-3}$) of the ground, resulting in weak influence of ground related reactions on atmospheric chemistry. Meanwhile, the simulated $O_3$ concentrations of both models are far higher





than the measured values, with calculated NMB values of 113.0 % and 113.8 %, respectively. The simulated $NO_2$

concentrations of both models also significantly exceed the measured values as seen in Fig. S6, with NMB values of 536.7 %

and 539.8 %, respectively. And the simulated HONO concentrations were far lower than the measured values, with the NMB

values being -94.3 % and -94.8 %, respectively as seen in Fig. S7. These findings indicate that the atmospheric environment

is more complex than smog chambers, due to additional chemical reactions influencing $O_3$ formation. To obtain accurate $O_3$

formation sensitivity, the model requires further revision.

**3.5 Further revision of models for complex atmospheric environments**

An important sink of $NO_2$ is its heterogeneous reactions on ground surface and aerosol surface in the atmosphere (Liu et al.,

2019; Xuan et al., 2024). The fact that the SVR model underestimates the ground level reaction rate of $NO_2$ and neglects its

heterogeneous reactions on aerosol surface is likely the major reason for the overestimation of simulated $NO_2$ concentrations,

which in turn leads to the overestimation of simulated $O_3$ levels. Table 3 summarizes the equations for $NO_2$ heterogeneous

reactions and the calculation methods for their rate constants.

**Table 3: Parametrization of heterogeneous reactions in AtChem2-MCM.**

| Heterogeneous reaction | Rate constant | Uptake coefficient | Notes |
|---|---|---|---|
| $2NO_2$ + ground surface → HONO + $HNO_3$ | $k_{gn} = \dfrac{1}{8} \times \gamma_g \times v_{NO_2} \times \dfrac{1.7}{H}$ | $\gamma_g = 8\times10^{-6}$ | Adopted from Liu et al. (2019) |
| $2NO_2$ + ground surface + $hv$ → HONO | $k_{gd} = \dfrac{1}{8} \times \gamma_{gd} \times v_{NO_2} \times \dfrac{1.7}{H}$ $\times \dfrac{J_{NO_2}}{0.007s^{-1}}$ | $\gamma_{gd} = 6\times10^{-5}$ | Adopted from Liu et al. (2019) |
| $2NO_2$ + aerosol surface → HONO + $HNO_3$ | $k_{an} = \dfrac{1}{4} \times \gamma_a \times v_{NO_2} \times S_a$ | $\gamma_a = 8\times10^{-6}$ | Adopted from Liu et al. (2019) |
| $2NO_2$ + aerosol surface + $hv$ → HONO | $k_{ad} = \dfrac{1}{4} \times \gamma_{ad} \times v_{NO_2} \times S_a \times \dfrac{J_{NO_2}}{0.007s^{-1}}$ | $\gamma_{ad} = 1\times10^{-3}$ | Adopted from Liu et al. (2019) |
| $NO_2$ → product | $k_{NO_2} = 1.5\times10^{-4}$ s$^{-1}$ | | |

$\gamma_g$ and $\gamma_a$ denote the uptake coefficients of $NO_2$ on ground surface and aerosol surface, respectively, while $\gamma_{gd}$ and $\gamma_{ad}$ represent the photo-enhanced uptake coefficients of $NO_2$ under illuminated conditions for ground and aerosol surfaces, respectively.


As shown in Fig. 3, after incorporating these heterogeneous reactions of $NO_2$ into the model (Named SVR NO₂Het model),

the simulation performance for $O_3$ improved, with an NMB value of 84.1 %. And the simulation performance of $NO_2$ has also

been enhanced as shown in Fig. S6, with an NMB value of 436.7 %. However, the simulated values of HONO were far higher



than the measured ones as shown in Fig. S7. Significant discrepancies still exist between the simulated and measured results,

indicating that some unidentified sinks of $NO_2$ have yet to be accounted for. Building on this, a constant sink of $NO_2$ was

incorporated into the model (Named SVR $NO_2$sink model), leading to a substantial improvement in $O_3$ simulation performance

(Fig. 3), with a NMB value of -5.2 %. The simulation performance of $NO_2$ and HONO also demonstrated notable accuracy,

achieving NMB values of -13.3 % and -12.4 %, respectively as shown in Fig. S6 and S7. As presented in Fig. S8, the simulated

concentrations of OH and $HO_2$ radicals reached peak values of $1.20 \times 10^7$ and $1.23 \times 10^9$ molecules cm$^{-3}$ s$^{-1}$, respectively.

These magnitudes are comparable to observations and model results reported for Beijing in previous studies (Slater et al., 2021;

Chai et al., 2023). However, the $O_3$ simulated levels for 16–17 August were significantly higher than the observed values. This

discrepancy can be attributed to the prevailing westerly winds starting from the 15 August (Fig. S5), which led to a change in

air masses and a significant decrease in $NO_2$ concentrations in the atmosphere. This change was not captured by the zero

dimensional box model, resulting in a substantial overestimation of $O_3$ concentrations. In conclusion, the simulation results

confirm the existence of unidentified $NO_2$ sinks in the atmospheric environment. Zheng et al. (2024) have shown that the ionic

strength in aerosol liquid water can enhance the uptake coefficient of $NO_2$ on aerosol surface and the reaction rate constant for

$NO_2$ they adopted in the field simulation is approximately on the order of $10^{-6}$ s$^{-1}$. Furthermore, evidence from Chu et al. (2023)

validates the mechanism of photo-enhanced heterogeneous reactions of $NO_2$ on building surfaces and their simulation results

indicated that the reaction rate constant of $NO_2$ related to $N_2O_5$ was on the order of $10^{-6}$ s$^{-1}$. However, these $NO_2$ reaction rate

constants were significantly lower than the artificially high values adopted in our model (see Table 3), and to our knowledge,

no previous studies have reported the rate constant of such magnitude in atmospheric environment. The reactions associated

with $NO_2$ sinks under the complex atmospheric environments require further investigation.

**3.6 Impact of model revision on $O_3$ formation sensitivity in the atmosphere**

EKMA curves of $O_3$ formation in Daxing District of Beijing were obtained by designing multiple sets of reduction scenarios

under basic model and SVR $NO_2$sink model, as shown in Fig. 4. Given that biogenic sources of VOCs are difficult to control,

only reductions in concentrations of anthropogenic VOCs (AVOCs) were considered in designing VOCs reduction scenarios.

These results demonstrate that both models consistently indicated that $O_3$ formation in Daxing District of Beijing during

summer is more sensitive to VOCs than to $NO_x$, aligning with findings from previous studies conducted in Beijing (Chai et al.,

2023; Han et al., 2023). However, a comparative analysis of model simulations revealed significant discrepancies between the

basic model and the SVR $NO_2$sink model in predicting $O_3$ formation in Daxing District of Beijing. The simulated $O_3$ values

from the basic model are far higher than those from the SVR $NO_2$sink model. Furthermore, the ridge line slope derived from

the EKMA curve of the basic model (1:2.41) is lower than that from the SVR $NO_2$sink model (1:2.06). These findings

collectively indicated that the basic model not only overestimated actual ambient $O_3$ levels but also distorted the non-linear

relationship between $O_3$ and its precursors, potentially misleading formulation of emission reduction policies. In contrast, the



predictions of $O_3$ concentrations by SVR $NO_2$sink model are closer to observed values and, critically, its steeper ridge line

slope indicated that the sensitivity of $O_3$ to VOCs in Daxing has been enhanced. When NO was constrained and additional

$NO_2$ sinks were considered in the model, the proportion of NO in $NO_x$ would increase. Elevated NO levels reduce the relative

importance of NO in its reactions with $HO_2$ and $RO_2$ radicals, thereby increasing the significance of these radicals, which are

key oxidation products of VOCs. Therefore, adding extra $NO_2$ sinks in the model makes $O_3$ formation more sensitive to VOCs.

Considering the Class I ambient air quality standard for $O_3$ in China (1-hour average: ~80 ppb), the results from SVR $NO_2$sink

model demonstrated that meeting the $O_3$ target requires either roughly a 76 % reduction in $NO_x$ by itself, or about a 60 %

reduction in VOCs by itself. But when reducing both two pollutants in coordination, a lower reduction ratio (53 % for NOx

and 46 % for VOCs, respectively) can achieve compliance requirements. Simulated emission reduction scenarios reveal VOCs

control is a higher priority over $NO_x$ reduction in achieving $O_3$ abatement targets for Daxing. The above content underscores

the critical importance of incorporating atmospheric $NO_2$ sinks into the box model for formulating scientific policies on $O_3$

emission reduction. A deeper investigation into the dominant atmospheric sinks of $NO_2$ is fundamentally important.

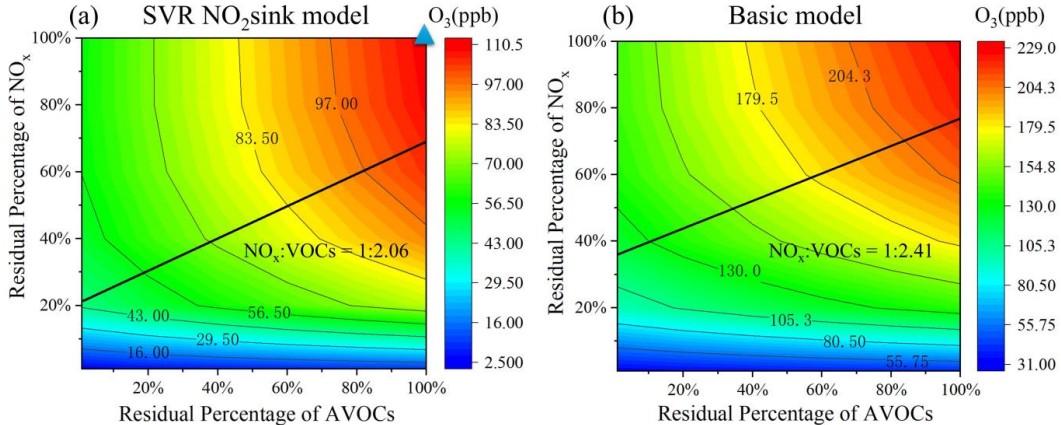

**Figure 4: Simulated $O_3$ EKMA curves based on (a) SVR NO₂sink model and (b) basic model in Daxing District during 13−19 August**

**2019. The triangular marker in the upper right corner of the (a) denotes the $O_3$ formation scenario before emission reduction.**

**4 Conclusions and implications**

Our study significantly improved the understanding of the response relationship between atmospheric $O_3$ and its precursors

through smog chamber experiments and box model revisions, and revealed the critical influence of the unidentified $NO_2$ sink

on $O_3$ sensitivity analysis. By incorporating reactions associated with the chamber wall into the box model, the simulation

performance for $O_3$ formation in smog chamber was markedly improved, with the average NMB reduced from −76.1 % to



−12.7 %. This improvement arises because such wall-related processes influence both the generation and elimination of OH radicals and the evolution of reactive nitrogen in the system, mainly including the release of OH radicals, the consumption of OH radicals by residual organics on the wall, and the heterogeneous reactions of $NO_2$, $N_2O_5$, and $HNO_3$. When applying the chamber-derived model to summer $O_3$ formation in Beijing's Daxing District, field-specific revisions were necessary. The most critical was adding an unidentified $NO_2$ sink. Incorporating this sink and other adjustments markedly improved performance, reducing NMB from 113.8 % to –5.2 % except for a few days influenced by air mass changes. Sensitivity analysis indicated an enhanced dependence of $O_3$ formation on VOC control in Daxing District. The introduction of the $NO_2$ sink accelerated $NO_2$ removal, weakened the scavenging effect of $NO_x$ on $HO_2$ and $RO_2$, thereby increasing radical concentrations and promoting VOC-driven $O_3$ production. This mechanism explains the shift in sensitivity regime, thereby highlighting the important regulatory role of $NO_2$ sink analysis in regional photochemical processes. This method can be extended to other regions following the steps illustrated in Figure 5, offering a scientific basis for developing regional $O_3$ pollution prevention and control strategies.

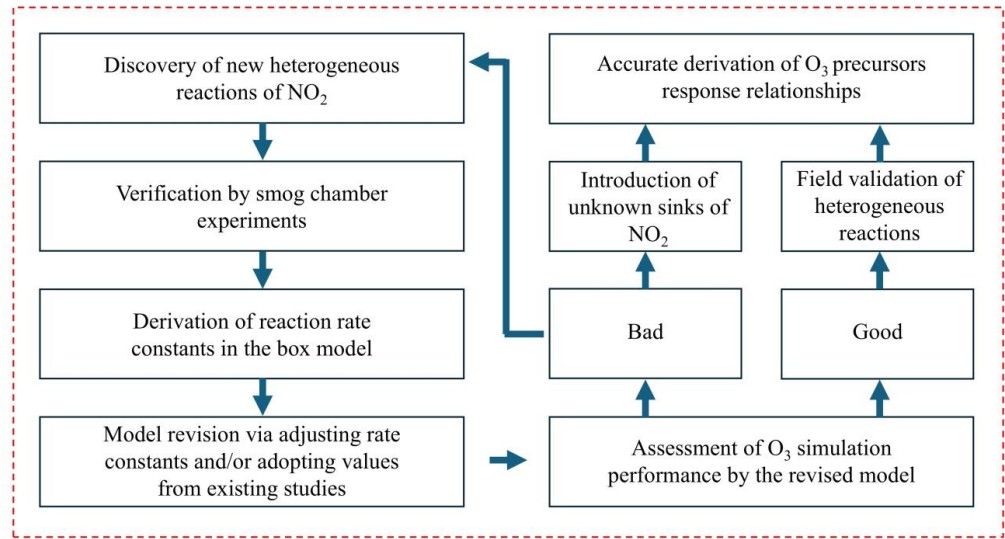

Figure 5: Schematic illustration of methods for accurately obtaining $O_3$ precursors response relationships.

Multiple studies employing different approaches have examined the sensitivity of $O_3$ formation in Beijing, consistently finding that $O_3$ production in the region is primarily VOC-limited (Cui et al., 2021; Nelson et al., 2024; Ji et al., 2025). This conclusion aligns with the diagnostic results of the present study, indirectly validating the reliability of the simulations. It should be noted, however, that the $NO_2$ sink used in this study exceeds levels reported in previous studies (Chu et al., 2023; Zheng et al., 2024). While this may lead to an overestimation of the magnitude of sensitivity changes, such a sink is still likely



to occur under complex atmospheric environment. Moreover, the zero-dimensional box model does not fully account for meteorological factors and does not consider the influence of regional transport, which may affect the applicability of the

results. Further systematic studies on $NO_2$ sinks are urgently needed to better simulate $O_3$ formation.

**Code and data availability.** The observation data at this site are available from the authors upon request.

**Supplement.** The supplement will be published alongside this article.


**Author contributions.** Conceptualization: BWC; Data curation: TZC, JL and HYX; Formal analysis: JLL; Funding acquisition: TZC and BWC; Investigation: JLL; Methodology: BWC, TZC and JLL; Resources: HH; Writing (original draft preparation): JLL and TZC; Writing (review and editing): TZC, BWC, JL, HYX, PZ, QXM, YHW and HL.

**Competing interests.** The authors declare that they have no conflict of interest.

**Acknowledgments.** The authors thank all the workers who provided support during this field observation period.

**Financial support.** This work was financially supported by the National Key Research and Development (R&D) Program of

China (2024YFC3714300), the National Natural Science Foundation of China (22376206 and 22188102), and the Youth Innovation Promotion Association, CAS (Y2022023 and Y2022021).

**Review statement.**

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
