# Peer review of "Response relationship between atmospheric O₃ and its precursors in Beijing based on smog chamber simulation and a revised MCM model"

_EGUsphere, 2025_

## Referee Comment (RC2)

This manuscript outlines attempts to improve the modelling of $O_3$ concentrations, and the production of NOx-VOC isopleths, by box modelling simulations. The paper has two halves. The first half describes a series of chamber experiments in which a chamber-specific mechanism is produced to improve $O_3$ concentration predictions. The second half attempts to use the developed chamber-specific mechanism to account for model biases in a box model of the ambient environment, suggesting that heterogeneous reactions could account for over-predictions in ozone concentrations from the base simulation.

The experimental procedures for the chamber experiments seem sound, and while there is some room for improvement in the explanation of the method by which the chamber-specific mechanism rates are selected (as explained later), the produced mechanism appears to be in line with existing literature and reproduces the observations well.

However, I believe that there are numerous issues with the implementation of the ambient box models, including the author's focus on $NO_2$ uptake as the sole explanation for model discrepancies. Some of these issues may be clarified with additional context in the explanation, but others may require deeper consideration. I have organised my comments into 4 sections: $NO_2$ Sink, General Comments, Specific Comments, and Minor Comments. I do believe there is value in efforts to improve $O_3$ predictions and that the potential role of $NO_2$ uptake is an important consideration, and I hope that addressing these comments can provide the improvements in the manuscript that I believe are necessary for publication.

**$NO_2$ Sink**

Generally, I feel that the entire manuscript fails to consider causes for the $O_3$ measurement discrepancy other than $NO_2$ uptake. The recommended workflow in Figure 5 completely omits the potential for processes (either chemical or physical) other than $NO_2$ uptake to influence $O_3$ concentrations. Given an initial over-prediction in $NO_2$, it would be possible to tune the $NO_2$ uptake rate to obtain 'accurate predictions' of any $O_3$ concentrations. However, this does not mean that the model is accurately representing the chemistry occurring, and the isopleth produced by such a model will not be accurate.

- At Line 300, the authors note that use of their $NO_2$sink model gives better $O_3$ predictions. This is only the case because the $NO_2$ sink has been tuned to give good $O_3$ predictions. Line 298 states that use of the Basic model distorts the produced isopleth, which is certainly true, but if the $NO_2$ sink model is "getting the right answer for the wrong reasons", then this isopleth would also be distorted in different ways.

- The authors do not provide any information on physical loss processes represented in the ambient box models. It is typical to include a first-order loss term for all species in the model (dilution term) to prevent the build-up of long-lived species. This term approximates physical removal of species in the real-world due to deposition and transport. Often a simple 24-hour loss lifetime is applied, though ideally a more detailed representation will be used that accounts for changing ventilation and deposition over time (e.g. Whalley et al. 2021 https://doi.org/10.5194/acp-21-2125-2021, Mayhew et al. 2022 https://doi.org/10.5194/acp-22-14783-2022 ).
  If the authors have not included such physical loss processes, then this is a major oversight that has the potential to explain the observed over-prediction.
  If the authors have included some representation of physical losses, then this needs to be explained in the manuscript. Additionally, the sensitivity of the $O_3$ concentration predictions to changes in physical losses should be assessed. This would provide evidence that the $O_3$ over-prediction needs to be explained chemically rather than simply being the result of the representation of physical processes. How does the $O_3$ concentration change if the physical loss rate is varied within the range of realistic values?
  In addition, if possible, measurements of VOC oxidation products could be compared to modelled concentrations to demonstrate that the model predicts those species well, especially once the large $NO_2$ sink is added. If the SVR $NO_2$ sink model predicts both VOC oxidation products and $O_3$ well, then this would lend further support to the presence of a large $NO_2$ sink.

- Related to the previous point, I believe that the authors' conclusions regarding the additional $NO_2$ loss term are far too strong, especially in light of the lack of investigation into the effect of physical processes on the conclusions. For example, Line 279 states that the model results "confirm the existence of unidentified $NO_2$ sinks". This is inappropriate given the large magnitude of the sink and the lack of physical mechanism offered by the authors. Similarly, Line 319 states that their model "revealed the critical influence of the unidentified NO2 sink".

  - Line 285: The authors state that heterogeneous $NO_2$ reactions are "likely the major reason" for over-predictions in $NO_2$. This statement requires justification, especially since the authors' own models demonstrate that adding heterogeneous $NO_2$ reactions does not correct the bias.

**General Comments**

- One of the central premises of the paper is to investigate the potential for non-HONO mediated ground reactions to impact atmospheric chemistry (Line 45 and 61). Figure 3 demonstrates that these reactions make an imperceptible difference to the model predictions, which the authors suggest is due to the small surface area-to-volume provided by the ground surface. This seems to be a good conclusion. Given that the introduction section states that the ground-mediated reactions are "systematically unassessed", the authors could highlight this conclusion more (e.g. in the conclusions section). This work seems to suggest that we have been right to ignore these surface processes in previous ambient box modelling studies (except maybe for the case of $NO_2$/HONO processes).

- Related to the previous point, I believe that it would be useful for the authors to illustrate how much of the additional $NO_2$ loss in the SVR $NO_2$Het model comes from the ground reaction vs. aerosol surface reactions.

- As with the representation of physical processes, the authors do not state whether a spin-up period was used for the simulation. The authors should note the length of the spin-up period used in the models, as is common practice.

- Is it true that all of the reactions from Table 2 are implemented in the ambient models (SVR model), with the rates scaled based on the ground surface area? If so, this assumes that the ground surface behaves chemically like the Teflon walls of the reaction chamber, which is a large assumption. The authors should highlight this assumption in the manuscript and discuss the ways that this might not be accurate. For example, Table 2 includes a reaction converting OH to $HO_2$ which is justified by the off-gassing of organics from the chamber walls. This off-gassing is unlikely to occur in the same way from non-plastic ground surfaces, and any VOCs produced in this manner would be measured during the campaign if they were of a high enough concentration to influence the chemistry.

   Since Figure 3 demonstrates that the inclusion of these ground surface processes makes no difference to the predicted $O_3$ concentrations, it is not of great importance for the authors to try to refine the wall reactions further, but the assumption of similar chemical behaviour between the chamber walls and the ground must be approached with an appropriate level of caution.

**Specific Comments**

- Line 87: The authors should explain why the fan was turned off during the experiments. It is my understanding that it is standard practice to leave fans running during experiments to ensure homogeneous mixing of the chamber air. For example, the authors acknowledge that wall processes influence the chemistry in the chamber. If the reaction mixture is not continuously mixed, could there not be potential for concentrations gradients to develop in the chamber air due to these wall processes? Alternatively, if poor mixing could occur as a result of air being removed/added at specific points in the chamber.

- Line 180: The authors note that they calculated the $O_3$ wall loss rate constant from experiments. They should explain in more detail how this was done. Did they perform experiments in the absence of VOCs and measure the decay rate of $O_3$? If so, it would be good to see the data from these experiments as supplementary information.

- Line 218: The authors should explain in greater detail why they are using Equation 3 to calculate the surface area to volume ratio, including units for the value of 1.7, explaining what this value denotes (increased surface area from surface roughness etc.), and why they believe it is appropriate to use the Li et al. 2010 value of 1.7 in their simulation (was it a similar ground-type?).

- Line 233: The authors note that they constrained the photolysis rate of $NO_2$ to measured values but do not state how (or if) they constrained photolysis rates of other species. AtChem2 has a JFAC feature that scales all photolysis rates based on a given measured photolysis rate, was this used here?

- Table 3: Is it the case that the final reaction in this table is the NO2 sink added only in the SVR $NO_2$sink model? If so, this is not clear as Line 259 mentions Table 3 before discussing the additional $NO_2$ uptake term. I would suggest either removing this entry from the table and stating the uptake rate constant in the main text (e.g. at line 270) or marking the $NO_2$ -> product reaction with an asterisk noting that it is only included in the SVR $NO_2$sink model. Either way, it should be made clear that this reaction is not in the SVR $NO_2$Het model.

- Line 278: The authors state that the over-predictions observed between the 16$^{th}$ and 17$^{th}$ of August are the result of a decrease in $NO_2$ during this period. Why is this not captured by the model considering NO is constrained?

- Line 301: The authors' discussion of how NO and $NO_2$ concentrations change when changing $NO_x$ seems plausible but given the availability of the data, the discussion should be accompanied with evidence from the model. For example, plots of NO to $NO_2$ ratio or $HO_2$ and $RO_2$ concentrations could be constructed.

**Minor Comments**

- Line 30: The authors should reference the WHO report that states the $O_3$ exposure limits, rather than linking to a web-page.
- Line 37: The authors state that $O_3$ formation occurs under "three distinct regimes". The use of the word distinct here is inaccurate. The presence of transitional regimes highlights that the boundary between $NO_x$-limited and VOC-limited regimes is not clear-cut (i.e. the regimes are not distinct).
- Line 39: The authors reference "the complexities of atmospheric conditions" that make modelling $O_3$ difficult. This is a vague statement, and the authors should clarify what they mean as there is difficulty in modelling many aspects of $O_3$ concentrations (emissions, meteorology, and atmospheric chemistry), while the focus of this work is only on chemistry.
- Line 96: The MCM is described as an "explicit mechanism", whereas it should be described as "semi-explicit" since it does perform lumping of some species, most notably in its treatment of $RO_2+RO_2$ reactions.
- Line 103: The correct reference for the AtChem2 description paper is Sommariva et al. 2020
- Line 224: It would be useful to highlight that while H denotes the boundary layer height, it is being used to calculate the volume of air present above 1 $m^2$ of surface. I.e. the units of H in Equation 3 are actually $m^3$, but the volume is equal to the height since the calculation is being performed for 1 $m^2$.
- Line 227: The values of 2.9E-3 and 5.8E-4 do not line up with the values obtained from Equation 3. I believe that this is just a typo since the correct value of 5.7E-3 is used further down at Line 247.
- Line 227: At the end of this paragraph, the authors mention a "further revised model", which I believe is the model they have been already discussing for the whole paragraph. I think it would be clearer if these final two sentences were moved to the start of the paragraph as an introduction to the section.
- Lines 231-239: Most of this paragraph is describing settings for the model, so should probably be included in the "Experimental Methods" section, as opposed to "Results and Discussion".
- Line 234: Again, I believe that the discussion of why $NO_2$ and HONO were not constrained would fit better in the "Experimental Methods" section.

---

## Author Comment (AC1)

**Responses to Reviewer #1**

Journal: Atmospheric Chemistry and Physics

Manuscript Number: egusphere-2025-3956

Title: Response relationship between atmospheric $O_3$ and its precursors in Beijing based on smog chamber simulation and a revised MCM model

We sincerely appreciate the your careful review and valuable guidance. The manuscript has been thoroughly revised according to the your suggestions, and all changes have been clearly highlighted using the Track Changes mode in the revised version. Enclosed please find our point-by-point responses to your comments for your kind consideration.

**Responses to your comments**

*This manuscript presents a well-structured and scientifically rigorous study that combines smog chamber experiments with a revised MCM box model to improve the simulation of $O_3$ formation and its sensitivity to precursors. The work is highly relevant to current air quality challenges, particularly in regions like Beijing suffering from severe $O_3$ pollution. The methodological approach is sound, the results are clearly presented, and the conclusions are well-supported by the data. The inclusion of chamber wall effects and unidentified $NO_2$ sinks represents a valuable contribution to the field. I recommend acceptance after minor revisions.*

**Response:** Thank you very much for your positive comments.

*General comments*

*1. The authors note that the $NO_2$ sink rate constant used is higher than values reported in previous studies (Page 12, Line 285). A brief justification or speculation on why this might be the case (e.g., unaccounted surfaces, synergistic effects) would strengthen the discussion.*

**Response:** The high $NO_2$ sink rate constant is likely attributable to physical dilution processes, which were not accounted for in the previous study. Previously, we attempted to simultaneously use NO and $NO_2$ as constraints. Under that configuration, the impact of varying physical dilution rates on $O_3$ simulation results was indeed minimal (as illustrated in the Figure R1), which led us to overlook the significance of the physical dilution process in the earlier stages of our study. However, as discussed in the revised manuscript, to more accurately evaluate the influence of ground-related reactions on $O_3$ formation, we transitioned to a more scientific setup where only NO is constrained. Under this revised configuration, the physical dilution rate exhibits a significant impact on the $O_3$ simulation results, as detailed in the Figure R2 of the revised manuscript. We have realized that neglecting physical processes in ambient atmospheric simulations is inappropriate, as it was the primary cause for the previous discrepancies where simulated $NO_2$ and $O_3$ concentrations deviated significantly from observations. However, the challenges in accurately characterizing atmospheric physical processes prevent further investigation into $O_3$ simulation performance and $O_3$ sensitivity (EKMA). Consequently, the focus of this study was shifted toward a systematic evaluation of the impact of ground-related reactions on the formation of $O_3$ and HONO. We conclude that ground-mediated reactions exert a significant influence on HONO, whereas their impact on $O_3$ is negligible. Detailed modifications have been implemented after line 296.

[Figure]

Figure R1. Comparison of observed and simulated $O_3$ concentrations across different model scenarios. The

basic model results are obtained with NO and $NO_2$ concentrations constrained. The dilute model results

reflect the impact of a 24 h physical dilution process on the simulated $O_3$ levels.

[Figure]

Figure R2. Comparison of observed and simulated $O_3$ concentrations under physical dilution rates of (a)

$5.80 \times 10^{-6}$ $s^{-1}$, (b) $1.16 \times 10^{-5}$ $s^{-1}$ and (c) $2.32 \times 10^{-5}$ $s^{-1}$. The black curves represent the observed $O_3$

concentrations, and the shaded areas denote nighttime periods. Compared to the basic model, the SVR

model incorporates ground-related reaction mechanisms derived from chamber experiments. The SVR

$NO_2$Het model further adjusts the ground-related $NO_2$ heterogeneous reactions based on the SVR model.

In the SVR $NO_2$Het 10 model, the rates of all reactions, excluding the $NO_2$ heterogeneous reactions, are

scaled up by a factor of 10 relative to the SVR $NO_2$Het model.

*2. The authors should briefly discuss the potential uncertainties in the uptake coefficients (γ) used for aerosols. Could the use of a constant γ, which may vary with aerosol composition and phase state, partly explain the need for such a large additional sink? A sentence or two on these limitations would be helpful.*

**Response:** The nocturnal aerosol uptake coefficient varies within the range of $2 \times 10^{-7}$ to $1 \times 10^{-5}$, and we adopted a relatively high constant value of $8 \times 10^{-6}$ (Liu et al., 2019). Similarly, a high uptake coefficient was applied for daytime conditions, with a peak value of $1 \times 10^{-3}$ (Wong et al., 2013). However, despite moderate improvements in $O_3$ simulation performance, a substantial discrepancy between modeled and observed values persists. This prompted us to explore alternative explanations, leading to the conclusion that the significant model-measurement gap is primarily driven by the neglect of physical processes.

*3. The discussion of the sensitivity shift (Section 3.6) could be enhanced by more explicitly linking the increased radical concentrations to the enhanced VOC sensitivity. A concise explanation could be: "The introduction of the $NO_2$ sink reduces $NO_2$ levels, which in turn lowers NO concentrations due to photo-stationary state relationships. Lower NO levels diminish the titration of $O_3$ and, more importantly, reduce the scavenging of $HO_2$ and $RO_2$ radicals by NO. This increases the radical chain length and amplifies the role of VOC oxidation in $O_3$ production, thereby shifting the system towards greater VOC sensitivity."*

**Response:** We are very grateful for your insightful suggestion, which has significantly enhanced our understanding of $O_3$ formation chemistry. We originally intended to simulate radical concentrations using our model to validate the theory you proposed. We acknowledge that the exact magnitudes of $NO_2$ sinks and physical processes are difficult to determine with the available data. Therefore, we have opted to omit a comprehensive discussion on the sensitivity transition to avoid over-interpretation.

*4. In the conclusion or discussion, it would be valuable to explicitly state that while the box model is excellent for isolating chemical mechanisms, the identified $NO_2$ sink rate constant is an "effective" rate that may also compensate for the lack of physical processes like advection and vertical dilution. A recommendation for future work using a 3D model with these revised chemical mechanisms to validate and spatially contextualize the findings would be a logical and strong ending point.*

**Response:** We fully agree that the $NO_2$ sink identified in our previous model effectively functioned as a surrogate to compensate for physical processes such as advection and vertical dilution. However, whether this compensatory effect introduces additional uncertainties warrants further investigation. Our findings highlight that physical processes exert a substantial influence on $O_3$ simulations. The robust capability of 3D models to resolve these physical dynamics facilitates a more rigorous investigation into the chemical mechanisms driving $O_3$ formation.

Revised text as it appears in line 359-360 of the text:

This underscores the necessity of employing three-dimensional models to further explore the complexities of $O_3$ chemistry.

*Technical comments*

*1. "the complex of atmospheric conditions" (Page 2, Line 39) might be better expressed as "the complexity of atmospheric conditions".*

**Response:** It has been revised.

Revised text as it appears in line 38-40 of the text:

However, the complexity of atmospheric chemical processes poses challenges for accurate characterization, resulting in significant biases in sensitivity analysis of $O_3$ formation (Li et al., 2018; Xue et

al., 2021; Qu et al., 2021; Chen et al., 2024), and triggering debates over optimal precursor control strategies.

*2. Page 3, Table 1, NO$_x$ concentration values are lower than those of NO, resulting in a negative NO$_2$ Correction should be applied to NO$_x$ concentrations (including both NO and NO$_2$). Additionally, the symbol "–" can be used to indicate that a reactant was not added to the chamber.*

**Response:** The instances in Table 1 where NO$_x$ appeared lower than NO was due to measurement uncertainties of the instruments. Since NO$_2$ was not intentionally added to the smog chamber in these cases, any measured NO$_2$ values that fell below zero were set to 0, and the initial NO$_x$ values were adjusted accordingly. Furthermore, we have changed the symbol used to denote components that were not added from "0" to "–" to ensure better clarity.

*3. Page 6, Line 151: change "revised model in experiment Iso&Tol02 were" to "revised model in experiment Iso&Tol02 are" for grammatical agreement.*

**Response:** It has been revised.

Revised text as it appears in line 174-175 of the text:

The NMB values for O$_3$ simulated by the basic model, revised model in experiment Iso&Tol02 are -83.9 % and -19.0 %, respectively.

*4. Page 7, Table 2: "Refered" should be "Referred".*

**Response:** It has been revised.

*5. "the slope of the ridge line of the EKMA curves" (Page 9, Line 203) is correct, but consider using*

*"ridgeline" as one word for consistency.*

**Response:** It has been revised.

Revised text as it appears in line 235-236 of the text:

Meanwhile, it can be found that, whether in the toluene only system or in the toluene isoprene mixed system, the slope of the ridgeline of the EKMA curves derived from the revised model changes.

*6. "the uptake coefficient in the chamber wall is equal to that in the atmospheric ground" (Page 9, Lines 218-219) – consider rephrasing to "on the chamber wall" and "on the ground surface" for precision.*

**Response:** It has been revised.

Revised text as it appears in line 254-257 of the text:

Given the complexity of surface types in the ambient environment and the significant variability in uptake coefficients across different surfaces (Vandenboer et al., 2013; Liu et al., 2019; Trick, 2004), we initially assumed that the uptake coefficient on the chamber wall is equivalent to that on the ground.

**References**

Chen, G., Xu, L., Yu, S., Xue, L., Lin, Z., Yang, C., Ji, X., Fan, X., Tham, Y. J., Wang, H., Hong, Y., Li, M., Seinfeld, J. H., and Chen, J.: Increasing Contribution of Chlorine Chemistry to Wintertime Ozone Formation Promoted by Enhanced Nitrogen Chemistry, Environmental Science & Technology, 58, 22714-22721, 10.1021/acs.est.4c09523, 2024.

Li, Q., Zhang, L., Wang, T., Wang, Z., Fu, X., and Zhang, Q.: "New" Reactive Nitrogen Chemistry Reshapes the Relationship of Ozone to Its Precursors, Environmental Science & Technology, 52, 2810-2818, 10.1021/acs.est.7b05771, 2018.

Liu, Y., Lu, K., Li, X., Dong, H., Tan, Z., Wang, H., Zou, Q., Wu, Y., Zeng, L., Hu, M., Min, K.-E., Kecorius, S., Wiedensohler, A., and Zhang, Y.: A Comprehensive Model Test of the HONO Sources Constrained to Field Measurements at Rural North China Plain, Environmental Science & Technology, 53, 3517-3525, 10.1021/acs.est.8b06367, 2019.

Qu, H., Wang, Y., Zhang, R., Liu, X., Huey, L. G., Sjostedt, S., Zeng, L., Lu, K., Wu, Y., Shao, M., Hu, M., Tan, Z., Fuchs, H., Broch, S., Wahner, A., Zhu, T., and Zhang, Y.: Chemical Production of Oxygenated Volatile Organic Compounds Strongly Enhances Boundary-Layer Oxidation Chemistry and Ozone Production, Environmental Science & Technology, 55, 13718-13727, 10.1021/acs.est.1c04489, 2021.

Trick, S.: Formation of Nitrous Acid on Urban Surfaces: a physical-chemical perspective, Ruperto Carola University Heidelberg, Heidelberg, Germany, 2004.

VandenBoer, T. C., Brown, S. S., Murphy, J. G., Keene, W. C., Young, C. J., Pszenny, A. A. P., Kim, S., Warneke, C., de Gouw, J. A., Maben, J. R., Wagner, N. L., Riedel, T. P., Thornton, J. A., Wolfe, D. E., Dubé, W. P., Öztürk, F., Brock, C. A., Grossberg, N., Lefer, B., Lerner, B., Middlebrook, A. M., and Roberts, J. M.: Understanding the role of the ground surface in HONO vertical structure: High resolution vertical profiles during NACHTT-11, Journal of Geophysical Research: Atmospheres, 118, 10,155-110,171, https://doi.org/10.1002/jgrd.50721, 2013.

Wong, K. W., Tsai, C., Lefer, B., Grossberg, N., and Stutz, J.: Modeling of daytime HONO vertical gradients during SHARP 2009, Atmos. Chem. Phys., 13, 3587-3601, 10.5194/acp-13-3587-2013, 2013.

Xue, M., Ma, J., Tang, G., Tong, S., Hu, B., Zhang, X., Li, X., and Wang, Y.: ROx Budgets and O3 Formation during Summertime at Xianghe Suburban Site in the North China Plain, Advances in Atmospheric Sciences, 38, 1209-1222, 10.1007/s00376-021-0327-4, 2021.

---

## Author Comment (AC3)

**Responses to Reviewer #3**

Journal: Atmospheric Chemistry and Physics

Manuscript Number: egusphere-2025-3956

Title: Response relationship between atmospheric $O_3$ and its precursors in Beijing based on smog chamber simulation and a revised MCM model

We sincerely appreciate the your careful review and valuable guidance. The manuscript has been thoroughly revised according to your suggestions, and all changes have been clearly highlighted using the Track Changes mode in the revised version. Enclosed please find our point-by-point responses to your comments for your kind consideration.

**Responses to your comments**

*This manuscript presents the relationship between $O_3$ and its precursors using smog chamber experiments and a revised MCM box model. The authors improve $O_3$ simulation by accounting for chamber wall effects under experimental conditions and unidentified $NO_2$ sinks under ambient conditions, highlighting the sensitivity of $O_3$ formation to VOCs and the implications for mitigation strategies in Daxing, Beijing.*

*However, the mechanisms related to chamber wall effects appear to have negligible influence when applied to real atmospheric conditions, whereas the unidentified NO2 sinks required for model-measurement agreement are unrealistically large. As a result, the revisions offer limited insights for ambient applications. Moreover, the simplified mechanisms used in the box model do not adequately represent interactions with meteorology or emissions. Therefore, I do not recommend the publication of this paper in ACP.*

**Response:** We appreciate your meticulous evaluation. The impact of ground-mediated reactions on $O_3$ has been sparsely documented in previous literature. Although our results indicate a minimal influence, we

contend that this finding is scientifically significant as it characterizes a previously unrecognized aspect of $O_3$. 1D models often encounter substantial uncertainties when resolving complex meteorological dynamics and emission profiles. Given the vast extent of the Earth's surface, the conclusions derived from our model are of significant reference value. Regarding the $NO_2$ sink, the following revisions have been implemented. Previously, we attempted to simultaneously use NO and $NO_2$ as constraints. Under that configuration, the impact of varying physical dilution rates on $O_3$ simulation results was indeed minimal (as illustrated in the Figure R1), which led us to overlook the significance of the physical dilution process in the earlier stages of our study. However, as discussed in the revised manuscript, to more accurately evaluate the influence of ground-related reactions on $O_3$ formation, we transitioned to a more scientific setup where only NO is constrained. Under this revised configuration, the physical dilution rate exhibits a significant impact on the $O_3$ simulation results, as detailed in the Figure R2 of the manuscript. However, due to the lack of glyoxal and boundary layer data, a simplified 24-h loss lifetime was employed to evaluate the sensitivity (Figure R2). Our results indicate that while physical processes exert a profound influence on the simulated $O_3$, the impact of ground-mediated reactions is negligible (Figure R2). We have realized that neglecting physical processes in ambient atmospheric simulations is inappropriate, as it was the primary cause for the previous discrepancies where simulated $NO_2$ and $O_3$ concentrations deviated significantly from observations. However, the challenges in accurately characterizing atmospheric physical processes prevent further investigation into $O_3$ simulation performance, VOCs simulation performance and $O_3$ sensitivity (EKMA). Consequently, the focus of this study was shifted toward a systematic evaluation of the impact of ground-related reactions on the formation of $O_3$ and HONO. We conclude that ground-mediated reactions exert a significant influence on HONO, whereas their impact on $O_3$ is negligible. Detailed modifications have been implemented after line 296.

[Figure]

Figure R1. Comparison of observed and simulated $O_3$ concentrations across different model scenarios. The base model results were obtained with NO and $NO_2$ concentrations constrained. The dilute model results reflect the impact of a 24 h physical dilution process on the simulated $O_3$ levels.

[Figure]

Figure R2. Comparison of observed and simulated $O_3$ concentrations under physical dilution rates of (a)

$5.80 \times 10^{-6}$ s$^{-1}$, (b) $1.16 \times 10^{-5}$ s$^{-1}$ and (c) $2.32 \times 10^{-5}$ s$^{-1}$. The black curves represent the observed O$_3$

concentrations, and the shaded areas denote nighttime periods. Compared to the basic model, the SVR

model incorporates ground-related reaction mechanisms derived from chamber experiments. The SVR

NO$_2$Het model further adjusts the ground-related NO$_2$ heterogeneous reactions based on the SVR model.

In the SVR NO$_2$Het 10 model, the rates of all reactions, excluding the NO$_2$ heterogeneous reactions, are

scaled up by a factor of 10 relative to the SVR NO$_2$Het model.

*Major Comments*

*1. The revised model (O$_3$ SVR) shows better agreement with chamber results than the base model, primarily*

*due to the introduction of ·OH generation associated with chamber wall effects. While such a mechanism*

*may be justified within a chamber, there is minimal physical basis for applying this wall-induced radical*

*source to ambient atmospheric conditions. Ground-related reactions on atmospheric chemistry do not mimic*

*chamber wall reactions, and extending this mechanism to the atmosphere is inappropriate.*

**Response:** After a thorough reassessment, we entirely concur with your perspective that the direct

extrapolation of wall-induced OH radical sources from smog chambers to complex ambient atmospheric

conditions lacks a solid physical basis.

Revised text as it appears in line 267-269 of the text:

However, the source mechanisms of OH radicals and the conversion of ·OH to HO$_2$ appear to be chamberspecific. Incorporating these incomplete mechanisms into models for simulating ambient conditions is

inappropriate and lacks physical basis.

*2. Reaction rate constants associated with wall effects in Table 2 are key parameters, but their optimization*

*process is insufficiently described (Page 6, Line 159; Page 7, Line 162 and Line 167). For example, how the reported $J_{NO2}$ = 0.0015 ppbv $s^{-1}$ (range 0.00075-0.0030 ppbv $s^{-1}$) in Angove et al. is converted to 1.2 × $10^6$ molecule $cm^{-3}$ $s^{-1}$ in this study (reaction 1: hv + wall → ·OH).*

**Response:** At standard temperature and pressure, 1 ppb is equivalent to 2.46 × $10^{10}$ molecules $cm^{-3}$ which allows the unit of the reported rate to be converted to molecules $cm^{-3}$ $s^{-1}$. Additional details regarding the specific optimization process have been included in the revised manuscript.

Revised text as it appears in line 183-196 of the text:

To achieve optimal model performance for VOCs across all experimental cases, the OH radical source rate constant reported by Wang et al. (2014) was adjusted within an order of magnitude (Wang et al., 2014). Following iterative optimization, the final OH production rate constant was determined to be 1.2 × $10^6$ molecules $cm^{-3}$ $s^{-1}$. Meanwhile, the mechanism of light-induced release of $NO_2$ from the wall (Bloss et al., 2005; Carter and Lurmann, 1991) was introduced to address the issue of relatively low production of $NO_2$ in the simulation. The $NO_2$ source rate constant from Wang et al. (2014) was varied within an order of magnitude. Through iterative optimization, the optimal $NO_2$ release rate constant was determined to be 6 × $10^5$ molecules $cm^{-3}$ $s^{-1}$, which is intermediate between the values reported by Angove et al. (Hynes et al., 2005) and Wang et al. (Wang et al., 2014). In addition, Teflon wall can release small amounts of organic impurities, which will consume OH radicals and generate $HO_2$ radicals (Metzger et al., 2008). Therefore, the additional mechanism that converts OH radicals into $HO_2$ radicals was also introduced into the model. This mechanism can accelerate the consumption of NO and also compensating for the deficiency of the simulated $NO_2$ and $O_3$ concentration. The relevant rate constants from Carter and Lurmann (1991) were varied within an order of magnitude. Through iterative optimization, the optimal conversion rate constant of OH radicals to $HO_2$ radicals is determined to be 10 $s^{-1}$, which lies within the range mentioned by Carter and Lurmann (Carter and Lurmann, 1991).

*The optimized rate constants may be chamber-specific. A controlled validation experiment using a small, clean plastic chamber (e.g., 1 m³) under similar conditions is therefore recommended. Is the revised model, with these parameters, able to reproduce results from such test-chamber conditions?*

**Response:** We acknowledge that wall effects and their corresponding rate constants exhibit significant variations across different smog chambers, a phenomenon that has been well-documented in previous studies (Hynes et al., 2005; Metzger et al., 2008; Wang et al., 2014). Therefore, within our current research framework, we maintain that conducting iterative optimization for the wall effects specific to our smog chamber is a robust and effective approach to ensure the reliability of the model parameters.

*Upper and lower limits for relevant parameters should also be provided, and a sensitivity analysis is recommended to assess how uncertainties in each parameter affect model-measurement discrepancies.*

**Response:** It has been revised accordingly. The results of the sensitivity analysis are shown in Figure S5.

Revised text as it appears in line 220-225 of the text:

Furthermore, sensitivity analyses for individual wall-related reactions were conducted based on the Iso&Tol02 and Iso&Tol04 experiments (Fig. S5) to quantify the influence of parameter uncertainties on model performance. The results indicate that the OH radical source mechanism and the OH-to-$HO_2$ conversion pathway exert substantial influence on the simulated $O_3$ concentrations. In contrast, the $NO_2$ source mechanism, $NO_2$ heterogeneous reactions, and $O_3$ wall loss show moderate impacts, while the remaining reactions have limited effects on the $O_3$ simulation results.

[Figure]

Figure S5: Sensitivity of simulated maximum $O_3$ concentrations to variations in reaction rate constants for experiments Iso&Tol02 and Iso&Tol04. The percentages represent deviations from the base simulation using a model incorporated with wall effect mechanisms. Most reaction rates were scaled by factors of 2 and 0.5, while the $N_2O_5 + H_2O$ reaction rate was specifically adjusted by factors of 10 and 0.1.

*1. Other experiments beyond Iso&Tol02 and Iso&Tol04 should be briefly described in the Supplement, including their experimental design and key results beyond simply displaying NMB values (Page 8, Line 185).*

**Response:** It has been revised accordingly.

Revised text as it appears in line 144-152 of the text:

To investigate the effect of precursor and chamber wall loss on $O_3$ formation, a series of experiments were conducted with varying precursor concentrations. In the mixed systems, results from Iso&Tol01, Iso&Tol02, and Tol01 demonstrated that $O_3$ production exhibited a decreasing trend as isoprene concentrations declined (Fig. 1, Fig. S4, and Table 1). Similarly, experiments Iso&Tol01, Iso&Tol03, and Iso01 showed that $O_3$

formation decreased with a reduction in toluene concentration (Fig. S4 and Table 1). For the Iso&Tol01, Iso&Tol04, and Iso&Tol05 series, $O_3$ production initially increased and subsequently decreased as $NO_x$ concentrations were reduced (Fig. 1, Fig. S4, and Table 1). In single VOC systems, experiments Iso01 and Iso02 indicated an increase in $O_3$ formation with higher isoprene concentrations. These observed relationships between $O_3$ and its precursors are consistent with established atmospheric chemistry theory, underscoring the reliability of our experimental results.

*2. The authors introduced a constant $NO_2$ sink to correct the model-measurement bias (Page 12, Line 270). However, the magnitude of this sink lacks physical justification and may simply force agreement for the wrong reasons. The authors should discuss alternative explanations for the observed bias beyond invoking an $NO_2$ uptake, and evaluate whether other processes could plausibly account for the discrepancy.*

**Response:** As previously stated, physical processes exert a substantial influence on the simulation results. These processes significantly modulate $O_3$ formation, thereby partially accounting for the systematic overestimation of $O_3$ and $NO_2$ relative to observations in earlier modeling studies.

*3. The attribution of $O_3$ overestimation in the SVR $NO_2$-sink model on 16-17 Aug to prevailing winds and associated air mass changes (Page 12, Line 277) requires further validation. The model overestimates $O_3$ continuously from late afternoon on Aug 15 through early morning Aug 16, during which wind speed decreased by more than a factor of four. Statistical analysis is needed to support the proposed explanation.*

**Response:** The mean wind speed during 15–17 was 2.31 m/s, whereas it averaged 1.76 m/s during other periods. We attribute the significant decrease in $NO_2$ concentrations during the night to air mass replacement events that occurred over these three days. Since 0-D box models are inherently unable to capture transport

processes such as air mass changes, the model could not predict this sharp decrease in $NO_2$. Consequently, this led to an overestimation of daytime $O_3$ concentrations in our simulations.

*Minor Comments*

*1. Subtitles for each section should be more concise.*

**Response:** It has been revised accordingly.

   The new subtitles are as follows:

   Line 143: 3.1 Construction of a revised model

   Line 211: 3.2 Evaluation of the revised model

   Line 226: 3.3 Impact of the revised model on the sensitivity of $O_3$

   Line 244: 3.4 Ambient application of the revised model

   Line 296: 3.5 Further revision of the model

*2. Table 1. Please explain why $NO_{x,0}$ is even lower than $NO_0$ under some experimental conditions?*

**Response:** These initial concentrations were measured using the Thermo 42i analyzer. In some cases, when $NO_2$ concentrations were below the detection limit, the instrument recorded negative values, which resulted in the calculated $NO_x$ concentrations being lower than NO. We have corrected the $NO_x$ concentrations in Table 1 to address this issue.

*3. Please clarify what J1-J56 correspond to in Table S1, and explain why J4 is selected in AtChem2-MCM in Table S2 (Supplement Page 6).*

**Response:** J1–J8 represent the photolysis rate constants for inorganic species, while J11–J56 correspond to

those for organic species. Collectively, these parameters reflect the spectral characteristics of the light source. The detailed correspondence between these constants and specific chemical species has been further clarified in Table S1 of the Supplementary Material. For more comprehensive information, please refer to the MCM official website. Specifically, J4 refers to the photolysis rate constant of $NO_2$, which is a core parameter governing tropospheric $O_3$ formation. Given that J4 directly determines the chemical production rate of $O_3$, its accuracy is critical. In our model simulations, we derived a photolysis scaling factor, JFAC, by comparing measured J4 values with their theoretical values. All other photolysis constants (J1–J56) were then synchronized using this factor to ensure that the simulated environment closely matches the actual light conditions.

Table S1: Photolysis rate constants used in the MCM box model for simulating smog chamber experiments.

| J | Species | Value (s-1) | J | Species | Value (s-1) | J | Species | Value (s-1) |
|---|---------|-------------|---|---------|-------------|---|---------|-------------|
| J1 | O3 | 1.05E-05 | J2 | O3 | 2.52E-06 | J3 | H2O2 | 4.85E-06 |
| J4 | NO2 | 9.20E-03 | J5 | NO3 | 2.17E-05 | J6 | NO3 | 5.37E-05 |
| J7 | HONO | 2.03E-03 | J8 | HNO3 | 5.66E-07 | J11 | HCHO | 1.94E-06 |
| J12 | HCHO | 2.20E-06 | J13 | CH3CHO | 9.73E-07 | J14 | C2H5CHO | 1.61E-06 |
| J15 | C3H7CHO | 5.20E-07 | J16 | C3H7CHO | 2.48E-07 | J17 | IPRCHO | 2.31E-06 |
| J18 | MACR | 5.12E-07 | J19 | MACR | 5.12E-07 | J20 | C5HPALD1 | 2.63E-04 |
| J21 | CH3COCH3 | 3.10E-07 | J22 | MEK | 2.81E-07 | J23 | MVK | 3.08E-07 |
| J24 | MVK | 3.08E-07 | J31 | GLYOX | 5.62E-07 | J32 | GLYOX | 5.17E-06 |
| J33 | GLYOX | 3.24E-05 | J34 | MGLYOX | 3.85E-05 | J35 | BIACET | 7.34E-05 |
| J41 | CH3OOH | 4.56E-07 | J51 | CH3NO3 | 9.22E-07 | J52 | C2H5NO3 | 1.76E-07 |
| J53 | NC3H7NO3 | 2.71E-04 | J54 | IC3H7NO3 | 2.86E-04 | J55 | TC4H9NO3 | 1.08E-06 |
| J56 | NOA | 7.26E-06 | | | | | | |

*4. Units should be provided for all variables in equations (1) and (2).*

**Response:** It has been revised accordingly.

In the Eq. (1), the actinic flux I (photons $cm^{-2}$ $s^{-1}$ $nm^{-1}$) is used to characterize the distribution of light intensity within the smog chamber. The absorption cross section σ ($cm^2$ $molecule^{-1}$) and the quantum yield Φ (dimensionless) describe the molecular light absorbing properties and energy conversion efficiency, respectively, during photolysis. And lambda is the wavelength (nm).

where $k$ ($s^{-1}$) is defined as the surface reaction rate constant for HONO, $\gamma$ (dimensionless) represents the uptake coefficient, $v_{NO_2}$ ($m \cdot s^{-1}$) denotes the average molecular velocity of $NO_2$, and $S/V_g$ ($m^{-1}$) stands for the surface-to-volume ratio of the ground.

*5. Please discuss whether effects other than wall loss may contribute to model-measurement discrepancies (Page 6, Line 144).*

**Response:** Thanks for your valuable suggestion. You are correct to highlight that discrepancies between model simulations and measurements are often multi-factorial. While we focused on analyzing the role of wall loss as a specific, often dominant factor in chamber-influenced scenarios, we acknowledge that a combination of the factors, such as the uncertainties in reaction rate constants, product yields, and the representation of heterogeneous or multiphase processes, contributes to the model-measurement discrepancies. It has been revised accordingly.

Revised text as it appears in line 166-169 of the text:

While we focused on analyzing the role of wall loss as a specific, often dominant factor in chamber-influenced scenarios, we acknowledge that a combination of the factors, such as the uncertainties in reaction rate constants, product yields, and the representation of heterogeneous or multiphase processes, contributes to the model-measurement discrepancies.

*6. Figure 1. Please explain why, despite the large base-model/measurement differences for both Iso&Tol02 and Iso&Tol04 throughout the process, the base-model/measurement difference at 6 h is much smaller for Iso&Tol04 than for Iso&Tol02?*

**Response:** The differences in simulation bias illustrated in Figure 1 primarily arise from the significantly distinct initial NO concentrations between the two experimental cases. In Iso&Tol02, the extremely high initial NO concentration led to intense $O_3$ titration reactions in the basic model. Furthermore, the wall effect leads to an increase in actual $O_3$ concentrations, a phenomenon that the basic model fails to capture. Both phenomena collectively lead to the significant discrepancy between the simulated and observed $O_3$ concentrations in Iso&Tol02. In contrast, the NO concentration in Iso&Tol04 was lower, and the titration effect was relatively weaker; thus, the simulation results from the basic model were more consistent with the observations.

*7. Figure 2 and Figure 4. Using a consistent colorbar range across subplots is recommended to improve comparability.*

**Response:** It has been revised accordingly.

[Figure]

Figure 2: Simulated EKMA curves for $O_3$ generation under the (a) toluene only system and (b) mixed VOCs system using the basic model. Correspondingly, (c) and (d) display the simulated EKMA curves for $O_3$ generation under the toluene only system and mixed VOCs system, respectively using the revised model.

*8. Table 3. Please clarify whether 0.007 s⁻¹ represents $J_{NO2}$ at noon and provide full definitions for $k_{gn}$, $k_{an}$, $S_a$, and other variables.*

**Response:** Regarding the parameter settings in Table 3, we have updated the original value from 0.007 s⁻¹ to 0.005 s⁻¹, which is a representative value informed by the literature (Liu et al., 2019).

Revised text as it appears in line 317-319 of the text:

$\gamma_g$ denote the uptake coefficients of $NO_2$ on ground surface, while $\gamma_{gd}$ represent the photo-enhanced

uptake coefficients of $NO_2$ under illuminated conditions for ground. $k_{gn}$ and $k_{gd}$ represent the first-order rate constants for the heterogeneous reaction of $NO_2$ during nighttime and daytime, respectively.

*Technical Corrections*

*1. Grammar issues should be corrected, including "the complex of atmospheric conditions" (Page 2, Line 39), "studying how secondary pollutants like $O_3$ formation" (Page 2, Line 48), and "details information" (Page 5, Line 116).*

**Response:** It has been revised accordingly.

Revised text as it appears in line 38-40, 46-47, and 114-115 of the text:

However, the complexity of atmospheric chemical processes poses challenges for accurate characterization, resulting in significant biases in sensitivity analysis of $O_3$ formation (Li et al., 2018; Xue et al., 2021; Qu et al., 2021; Chen et al., 2024), and triggering debates over optimal precursor control strategies.

Smog chamber has emerged as an indispensable approach for studying the formation of secondary pollutants such as $O_3$ (Chen et al., 2022; Carter et al., 1995).

[revised manuscript text omitted]